# Mechanism of U6 snRNA oligouridylation by human TUT1

Seisuke Yamashita ⬚[1] & Kozo Tomita ⬚[1] ✉

U6 snRNA is a catalytic RNA responsible for pre-mRNA splicing reactions and undergoes various post-transcriptional modifications during its maturation process. The 3'-oligouridylation of U6 snRNA by the terminal uridylyl-transferase, TUT1, provides the Lsm-binding site in U6 snRNA for U4/U6 di-snRNP formation and this ensures pre-mRNA splicing. Here, we present the crystal structure of human TUT1 (hTUT1) complexed with U6 snRNA, representing the post-uridylation of U6 snRNA by hTUT1. The N-terminal ZF-RRM and catalytic palm clamp the single-stranded AUA motif between the 5'-short stem and the 3'-telestem of U6 snRNA, and the ZF-RRM specifically recognizes the AUA motif. The ZF and the fingers hold the telestem, and the 3'-end of U6 snRNA is placed in the catalytic pocket of the palm for oligouridylation. The oligouridylation of U6 snRNA depends on the internal four-adenosine tract in the 5'-part of the telestem of U6 snRNA, and hTUT1 adds uridines until the internal adenosine tract can form base-pairs with the 3'-oligouridine tract. Together, the recognition of the specific structure and sequence of U6 snRNA by the multi-domain TUT1 protein and the intrinsic sequence and structure of U6 snRNA ensure the oligouridylation of U6 snRNA.

In eukaryotes, the pre-mRNA splicing reaction is catalyzed by the spliceosome, a ribonucleoprotein (RNP) complex. The major spliceosome comprises five small RNP complexes—U1, U2, U4, U5, and U6 snRNPs—and a large number of proteins[1]. The U6 snRNP enters the splicing cycle by forming di-U4/U6 snRNPs in a reaction catalyzed by the p110/SART3 and Lsm2-8 ring protein complexes, which together promote the annealing of U6 and U4 snRNAs[2–4]. The di-U4/U6 snRNPs are used to form the U4/U6·U5 tri-snRNP, which is then recruited into the pre-spliceosome consisting of the pre-mRNA and the U1 and U2 snRNPs. The U6 snRNA forms an alternative helix with the U2 snRNA, and the splicing reactions proceed with the structural rearrangements of the U6 snRNA. In the splicing reactions, U6 snRNA catalyzes the trans-esterification splicing reactions by forming the active site with the coordinating divalent cations for the catalysis[5].

U6 snRNA is transcribed by RNA polymerase III. The U6 snRNA transcript has 5'-short stem and 3'-telestem-internal stem-loop (ISL) secondary structures, which are separated by a single-stranded region containing the AUA motif[6,7]. The U6 snRNA primary transcript undergoes multiple processing steps, including modifications of the nucleosides, 5'-γ-phosphate methylation, 3'-oligouridylation, and 3'-uridine trimming[8]. The primary transcript of the human U6 snRNA has four genome-encoded uridines (4 Us: $UUUU_{OH}$) at its 3'-end. After transcription, the 3'-end is oligouridylated by a terminal uridylyl-transferase (TUTase), TUT1 (TENT1)[9–11]. Subsequently, the 3'-oligouridylated tail is trimmed by Mpn1 (Usb1), a 3'–5' exonuclease[12–14]. The mature U6 snRNA has five uridines with a 2',3'-cyclic phosphate (5 U > p: UUUUU>p). These 3'-maturation processes protect U6 snRNA from degradation. The 3'-oligouridylated tail is required for the di-U4/U6 snRNP formation. The Lsm2-8 ring complexes bind the oligouridylated tail of U6 snRNA, and together with p110 facilitate the annealing of the U6 and U4 snRNAs to form the di-U4/U6 snRNP[4,15,16]. The 3'-oligouridylated tail is also required for the U6 snRNA recycling after the splicing reaction[17], and thus contributes to efficient pre-mRNA splicing in cells.

Human TUT1 (hTUT1) specifically oligouridylates U6 snRNA using UTP as a substrate[9–11], and is a member of the TENT family[18–20]. hTUT1 is

[1]Department of Computational Biology and Medical Sciences, Graduate School of Frontier Sciences, The University of Tokyo, Kashiwa, Chiba 277-8562, Japan. ✉e-mail: kozo-tomita@edu.k.u-tokyo.ac.jp

a multi-domain enzyme consisting of the N-terminal zinc finger (ZF), the RNA recognition motif (RRM), the central catalytic core domain, and the C-terminal kinase associated-1 (KA-1) domain[21–25]. The detailed mechanism for the nucleotide specificity of hTUT1 was clarified by structural and biochemical analyses of the catalytic core domain of hTUT1 complexed with UTP[22]. However, the detailed mechanisms for the specific recognition of U6 snRNA by the multiple domains of hTUT1 are not fully understood. Furthermore, the regulatory mechanism ensuring the oligouridylation of the 3'-end of U6 snRNA also remains enigmatic.

Here, we present the crystal structure of a shortened hTUT1 complexed with a short U6 snRNA, representing the post-uridine addition to the 3'-end of U6 snRNA by hTUT1. Crystallographic and biochemical studies of hTUT1 have now revealed the molecular mechanism underlying the specific oligouridylation of the structured U6 snRNA and the regulation of the 3'-oligouridylation of U6 snRNA by hTUT1.

## Results

### Crystallization of the TUT1-U6 snRNA complex
Human TUT1 (hTUT1) is a multi-domain protein consisting of the N-terminal ZF, RRM, catalytic core domains (palm and fingers), and the C-terminal KA-1 domain[23]. A proline-rich region (PRR), which is less conserved among vertebrate TUT1 proteins and dispensable for the oligouridylation of U6 snRNA in vitro[22], is present within the palm. The nuclear localization signal (NLS) is inserted within the C-terminal KA-1 domain (Fig. 1a, Supplementary Fig. 1).

The crystallization trials for the full-length hTUT1 (hTUT1_FL) complexed with an in vitro transcribed full-length U6 snRNA ending with four 3'-uridines (3'– 4 Us) were unsuccessful. Thus, we designed a shorter hTUT1 and a shorter U6 snRNA for the crystallization of the hTUT1-U6 snRNA complex, based on our previous biochemical and structural analyses of hTUT1[22], as described below. The C-terminal KA-1 domain of hTUT1 is bound around the stem of the ISL and the bulged region between the telestem and the ISL of U6 snRNA[22] (Fig. 1b). In addition, the N-terminal ZF and RRM are bound around the single-stranded region containing the AUA motif between the 5'-short stem and 3'-stem (telestem and ISL) (Fig. 1b). While the KA-1 deletion from hTUT1 decreased the U6 snRNA oligouridylation activity (-20% of hTUT1_FL) by reducing the hTUT1 affinity for U6 snRNA, the N-terminal ZF and RRM deletion almost abolished the U6 snRNA oligouridylation (-<0.2% of hTUT1_FL) by reducing the affinity for RNA as well as the catalysis in vitro[22]. Therefore, for a short hTUT1, we designed the hTUT1 lacking the PRR and the KA-1, hereafter termed hTUT1_ΔC (Fig. 1a). For a shorter U6 snRNA, we designed U6 snRNA lacking the 5'-short stem and the ISL and possessing four 3'-Us (U103-U106), hereafter termed U6_mini (Fig. 1b).

hTUT1_ΔC oligouridylates U6 snRNA ending with four 3'-Us (U103-U106) as efficiently as the hTUT1_FL and rapidly produces U6 snRNA with six 3'-Us (U103-U108) (Fig. 1c). hTUT1_ΔC also oligouridylates U6_mini ending with four 3'-Us and produces U6_mini with six 3'-Us, although the hTUT1_FL oligouridylates U6_mini and produces U6_mini with six 3'-Us more efficiency and rapidly than hTUT1_ΔC (Fig. 1c). For the crystallization, several cysteine residues in the hTUT1_ΔC were replaced with serine or alanine[22] (Supplementary Fig. 2) and one of the catalytic residues, aspartate (Asp218), was replaced with alanine (Supplementary Fig. 1).

The X-ray diffractive crystal of hTUT1_ΔC complexed with U6_mini was obtained, and its structure was analyzed. The crystal belongs to the space group $P6_522$ and contains one hTUT1_ΔC-U6_mini complex in the asymmetric unit cell. The initial phase was determined by the molecular replacement method, using the structures of the catalytic core domains of hTUT1 (PDB ID: 5WU1 and 5WU6)[22] as search models. Finally, the structure was refined to an $R_{work}$ of 30.2% ($R_{free}$ = 33.9%) at 3.7 Å resolution. The scaling and refinement statistics are summarized in Supplementary Table 1, and representative images of the electron density are shown in Supplementary Fig. 3.

### Overall structure of the hTUT1_ΔC-U6_mini complex
The structure of the hTUT1_ΔC-U6_mini complex revealed the extensive interactions between U6_mini and multiple domains of hTUT1_ΔC (Fig. 1d, e, Supplementary Figs. 4, 5). The linker region (amino acid residues 135–144) between the RRM and the catalytic core domain of hTUT1_ΔC was not visible in the present structure. The N-terminal ZF and the RRM constitute a single domain without any linker between them (Fig. 1d, e, Supplementary Fig. 5a), and the ZF-RRM and the palm of hTUT1_ΔC together clamp the 5'-single stranded region containing the AUA motif of U6_mini (Fig. 1d, e, Supplementary Fig. 5a−e). The ZF and the groove loop (amino acid residues 423–427) in the fingers hold the double-stranded telestem from the major and minor grooves of the stem region of U6_mini, respectively (Fig. 1d, Supplementary Fig. 5b, f, g), and the 5'-anchor (amino acid residues 357–371) of the palm stacks with the base-pair at the top of the telestem (Fig. 1d, Supplementary Fig. 5d). As a result, the 3'-end of U6_mini is placed into the catalytic cleft between the palm and fingers (Fig. 1d, e, Supplementary Fig. 5g).

The structure of the catalytic core domain, the palm and fingers, in the hTUT1_ΔC-U6_mini complex is virtually identical to those of the previously determined structures of hTUT1 lacking the ZF and KA-1 (hTUT1_ΔZFΔC, Fig. 2a, Supplementary Fig. 6a, b, PDB ID: 5WU6) and the hTUT1 lacking the ZF and RRM (hTUT1_ΔZF-RRM, Fig. 2b, Supplementary Fig. 6a, b, PDB ID: 5WU1).

The location of the RRM relative to the catalytic core domain in the structure of hTUT1_ΔZFΔC is different from that in the structure of the hTUT1_ΔC-U6_mini complex (Fig. 2c, d). The N-terminal ZF-RRM is connected to the catalytic domain via a flexible linker (Fig. 1d, e) and can be mobile relative to the catalytic core domain in the absence of an RNA substrate[22]. In the oligouridylation of U6 snRNA by hTUT1, the mobile N-terminal ZF-RRM binds the single-stranded AUA motif between the 5'-short stem and the telestem of U6 snRNA, and together with the palm, clamps the single-stranded region and pulls the telestem of U6 snRNA into the catalytic cleft of hTUT1 for 3'-oligouridylation (Figs. 1e, 2d, Supplementary Fig. 6c). As described below, the ZF-RRM specifically recognizes the single-stranded AUA motif (A22U23A24) in U6 snRNA. These structural features explain the tight and stable interaction between hTUT1 and U6 snRNA[22].

In the superimposition of the structure of the hTUT1_ΔC-U6_mini complex onto that of the hTUT1_ΔZF-RRM (Fig. 2e), the C-terminal KA-1 is close to the tip of the U6_mini loop in the hTUT1_ΔC-U6_mini complex (Fig. 2e), and the arginine-rich region (RRR) in KA-1 (Supplementary Fig. 1) could interact with the RNA. This structural model is consistent with previous biochemical data showing that the KA-1 of hTUT1 binds around the ISL and the bulged region between the telestem and ISL of U6 snRNA (Figs. 1b, 2e) and increases the affinity of hTUT1 for U6 snRNA through the RRR[22]. In the oligouridylation of U6 snRNA by hTUT1, the telestem and ISL of U6 snRNA are held by the ZF and the fingers and KA-1, respectively (Fig. 2e, f, Supplementary Fig. 6c). The KA-1 of hTUT1 would prevent the U6 snRNA from dislodging from the surface of hTUT1 for efficient oligouridylation. The exact RNA recognition mechanism by KA-1 awaits the future determination of the full-length hTUT1 complexed with U6 snRNA[21].

### Recognition of the single-stranded AUA motif
The overall structure of the hTUT1_ΔC-U6_mini complex suggests that hTUT1 recognizes specific structural features of U6 snRNA (Fig. 1, Supplementary Figs. 4, 5). Indeed, the substitutions of the amino acid residues that interact with U6 snRNA decreased the oligouridylation of U6 snRNA (Supplementary Fig. 7).

In particular, the single-stranded AUA motif between the 5'-short stem and the 3'-telestem of U6 snRNA is clamped by the ZF-RRM and

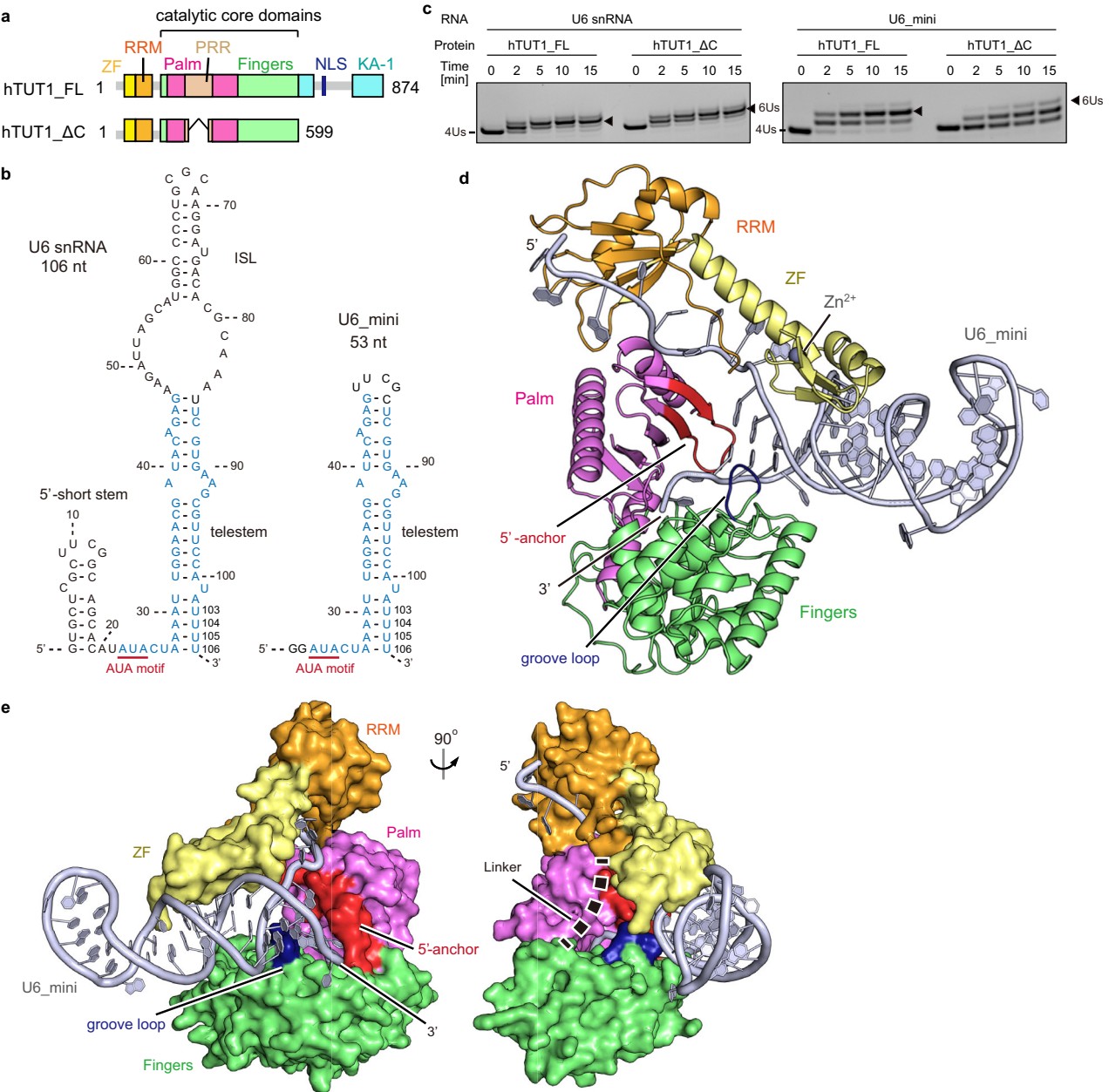

**Fig. 1 | Overall structure of the hTUT1_ΔC-U6_mini complex. a** Schematic diagrams of full-length human TUT1 (hTUT1_FL) and its variant lacking the proline-rich region (PRR) and the C-terminal kinase associated 1 (KA-1) domain (hTUT1_ΔC) used for crystallization. ZF (Zinc finger: yellow), RRM (RNA recognition motif: orange), PRR (brown), palm (magenta), fingers (green), KA-1 (cyan), and NLS (blue). **b** Secondary structures of human U6 snRNA and its variant U6_mini used for crystallization. Both U6 snRNA and U6_mini have four 3'-uridines (4 Us: U103U104U105U106). U6_mini lacks the 5'-short stem and internal stem-loop (ISL). The common nucleotide sequences in U6 snRNA and U6_mini are colored cyan. U6_mini has the 5'-GG sequence for in vitro transcription by T7 RNA polymerase. The ISL is replaced with the UUCG tetraloop, and U85 is replaced with C85. **c** In vitro uridylation of U6 snRNA (left) and U6_mini (right) by hTUT1_FL and hTUT1_ΔC. RNA (200 nM; U6 snRNA or U6_mini) was incubated with 20 nM hTUT1_FL (or hTUT1_ΔC) in the presence of 1 mM UTP. The arrowheads are reaction products with six uridines (6 Us) at their 3'-ends. The experiments were performed twice, and the representative gel images are shown. Source data are provided as a Source Data file. **d** The overall structure of the hTUT1_ΔC-U6_mini complex. ZF, RRM, palm, and fingers are colored as in (**a**), and U6_mini is colored gray. **e** Surface representation of the hTUT1_ΔC-U6_mini complex structure. ZF, RRM, palm, and fingers are colored as in (**d**). The 5'-anchor (residues 357-371) in the palm and the groove loop (residues 423-427) in the fingers are colored red and blue, respectively. The flexible linker (residues 135-144) between the RRM and the fingers is depicted as a dashed line.

the palm (Fig. 1d, e), and the A22U23A24 sequence is specifically recognized by the ZF-RRM through hydrogen-bond and stacking interactions (Fig. 3a). Asp90 and Lys91 form hydrogen bonds with the 6-NH2 group and the 7-N atom of A22, respectively. Arg124 and the main-chain carbonyl oxygen of Pro125 form hydrogen bonds with the 4-O and 3-N atoms of U23, respectively. The side-chains of Glu127 and Phe59 sandwich the U23 base. Gln53 and Arg126 form hydrogen bonds

with the 6-NH2 group and the 7-N atom of A24, respectively. Substitutions of Glu53, Asp90, and Arg126 with alanine all decreased the 3'-oligouridylation of U6 snRNA (Fig. 3b), while the Arg124Ala mutation did not affect the reaction. Since the 3-N atom of U23 interacts with the carbonyl oxygen main chain of Pro125, and the U23 base is stacked with the side-chains of Glu127 and Phe59, the effect of the Arg124Ala mutation would not be detected under the tested conditions.

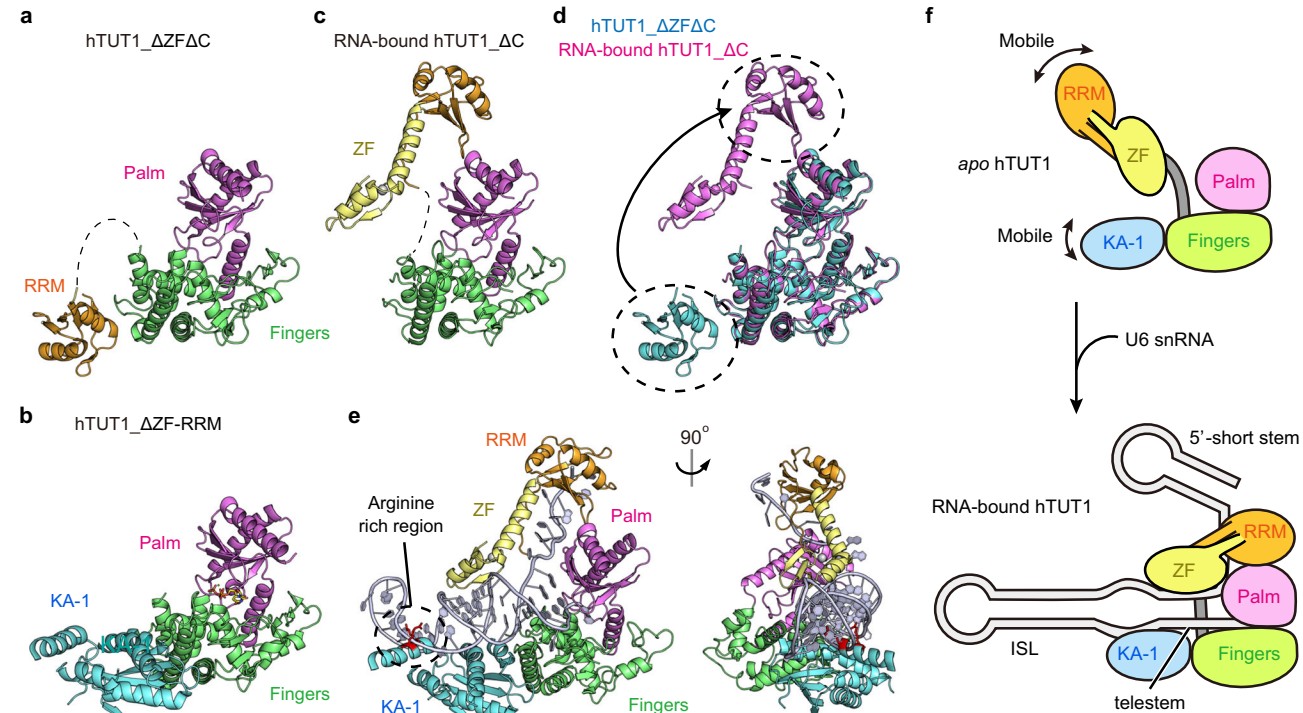

**Fig. 2 | The movement of N-terminal ZF-RRM upon hTUT1 binding to U6 snRNA. a** Structure of hTUT1 lacking the ZF and the KA-1 (hTUT1_ΔZFΔC, PDB ID: 5WU6)[22]. The domain colors are the same as in Fig. 1. The flexible linker between the RRM and the fingers (residues 127-144) is depicted as a dashed line. **b** Structure of hTUT1 lacking the ZF and RRM (hTUT1_ΔZF-RRM (PDB ID: 5WU1)). KA-1 is colored cyan. **c** Structure of hTUT1_ΔC in complex with U6_mini. For clarity, the U6_mini was omitted. The domain colors are the same as in Fig. 1. The flexible linker between the RRM and the fingers (residues 135-144) is depicted as a dashed line. **d** Superimposition of the hTUT1_ΔZFΔC (**a**) (cyan) and U6_mini-bound hTUT1_ΔC (**c**) (magenta). **e** A model of full-length hTUT1 complexed with U6_mini. The structure of hTUT1_ΔZF-RRM (**b**) was superimposed onto the hTUT1_ΔC-U6_mini complex. The arginine-rich region (RRR) in the KA-1 domain (cyan) is colored red. **f** A schematic representation of the domain movements upon hTUT1 binding to U6 snRNA. The domain colors are the same as in (**e**).

To assess the requirement of the AUA motif in U6 snRNA for efficient uridylation by hTUT1, the mutant RNA substrates were tested for uridylation by hTUT1. While the addition of the AU sequence to the 5' end of the A22U23A24 motif of U6_mini (U6_mini+2) did not increase the oligouridylation of the RNA by hTUT1_FL, the removal of the AU in the 5'-single-stranded AUA motif of U6_mini (U6_mini-2) decreased the oligouridylation (Fig. 3c, d). Based on gel-shift assays, the estimated $K_d$ values of the ZF-RRM for U6_mini+2, U6_mini (wild) and U6_mini-2 were $258 \pm 46$, $204 \pm 41$ and $574 \pm 78$ nM, respectively (Fig. 3e). Thus, the decreased oligouridylation of U6_mini-2 is due to a reduction in the affinity of ZF-RRM. Furthermore, the substitution of the A22U23A24 motif in the single-stranded region between the 5'-short stem and 3'-telestem of U6 snRNA with CCC (AUA/CCC) or GGG (AUA/GGG) also decreased the oligouridylation by hTUT1_FL (Fig. 3f, g). The gel shift assays provided estimated $K_d$ values of the ZF-RRM for U6 snRNA, U6_AUA/CCC and U6_AUA/GGG of $124 \pm 21$, $290 \pm 29$ and $258 \pm 20$ nM, respectively (Fig. 3h). The Q53A/R126A and F59A/F94A mutations in ZF-RMM decrease the affinity for U6 snRNA with A22U23A24 motif (Supplementary Fig. 8). Altogether, the recognition of the single-stranded A22U23A24 motif in U6 snRNA by hTUT1 is crucial for its efficient oligouridylation.

### Mechanism of U6 snRNA oligouridylation by hTUT1

In the structure of the hTUT1_ΔC-U6_mini complex, the 3'-terminal uridine of U6_mini, corresponding to U106 of U6 snRNA (Fig. 1b), resides in the incoming UTP-binding site (+1 position) in the catalytic pocket, and the uridine base stacks with the uridine base of U105 (−1 position, 3'-priming position) (Fig. 4a, b). The 4-O atom of U106 forms hydrogen bonds with the conserved histidine (His549) (Fig. 4a), similar to other TUTases that recognize UTP in the catalytic pocket[22,26–28]. Thus, the present structure represents the post-U-addition stage. In the structure

of the hTUT1_ΔC-U6_mini complex, the 5'-anchor (Gln359 and Val361) in the palm interacts with the bases at the top of the stem region, while the groove loop (amino acid residues 423 − 427) in the fingers interacts with the minor groove of the stem (Fig. 4a, b, Supplementary Fig. 5g). As a result, U105 (−1 position) is directed to the active site of hTUT1.

hTUT1_FL can efficiently uridylate U6 snRNAs ending with three to five 3'-uridines (3 Us: U103-U105, 4 Us: U103-U106, and 5 Us: U103-U107) to produce U6 snRNAs with six 3'-uridines (6 Us: U103-U108). However, hTUT1_FL cannot produce U6 snRNAs with more than seven uridines efficiently (Fig. 4c, Supplementary Fig. 9a).

It was assumed that the four-adenine tract (A27−A30) in the telestem of U6 snRNA could define the number of uridines added to the 3' end of U6 snRNA, regulate the uridine incorporation and ensure the oligouridylation. The oligouridylation proceeds until U103−U106 form a duplex with the four-adenine tract (A27−A30), and U107 and U108 occupy positions −1 and +1 in the catalytic pocket, respectively (Fig. 4d). In support of this assumption, the U6 snRNA mutant with four 3'-Us (U103−U105) and a three-adenine tract in the telestem (delA_4 Us) was oligouridylated efficiently to produce U6 snRNA with five 3'-Us (Fig. 4e, f, Supplementary Fig. 9b, c). The U6 snRNA mutant with four 3'-Us and a five-adenine tract in the telestem (insA_4 Us) was oligouridylated by hTUT1_FL to produce U6 snRNAs with seven 3'-Us (Fig. 4e, f, Supplementary Fig. 9c, d). Thus, the number of incorporated uridines is regulated by the four-adenine tract (A27−A30) in the telestem of U6 snRNA.

After the U108 incorporation, the telestem of U6 snRNA translocates, and U107 and U108 move to the −2 and −1 positions, respectively (Fig. 4d). U107 at the −2 position cannot base pair with U26. Thus, further uridylation rarely occurs, due to the improper geometry of U108 at the −1 position and the incoming UTP at the +1 position for the nucleotidyltransfer reaction (Fig. 4d). This would result in the

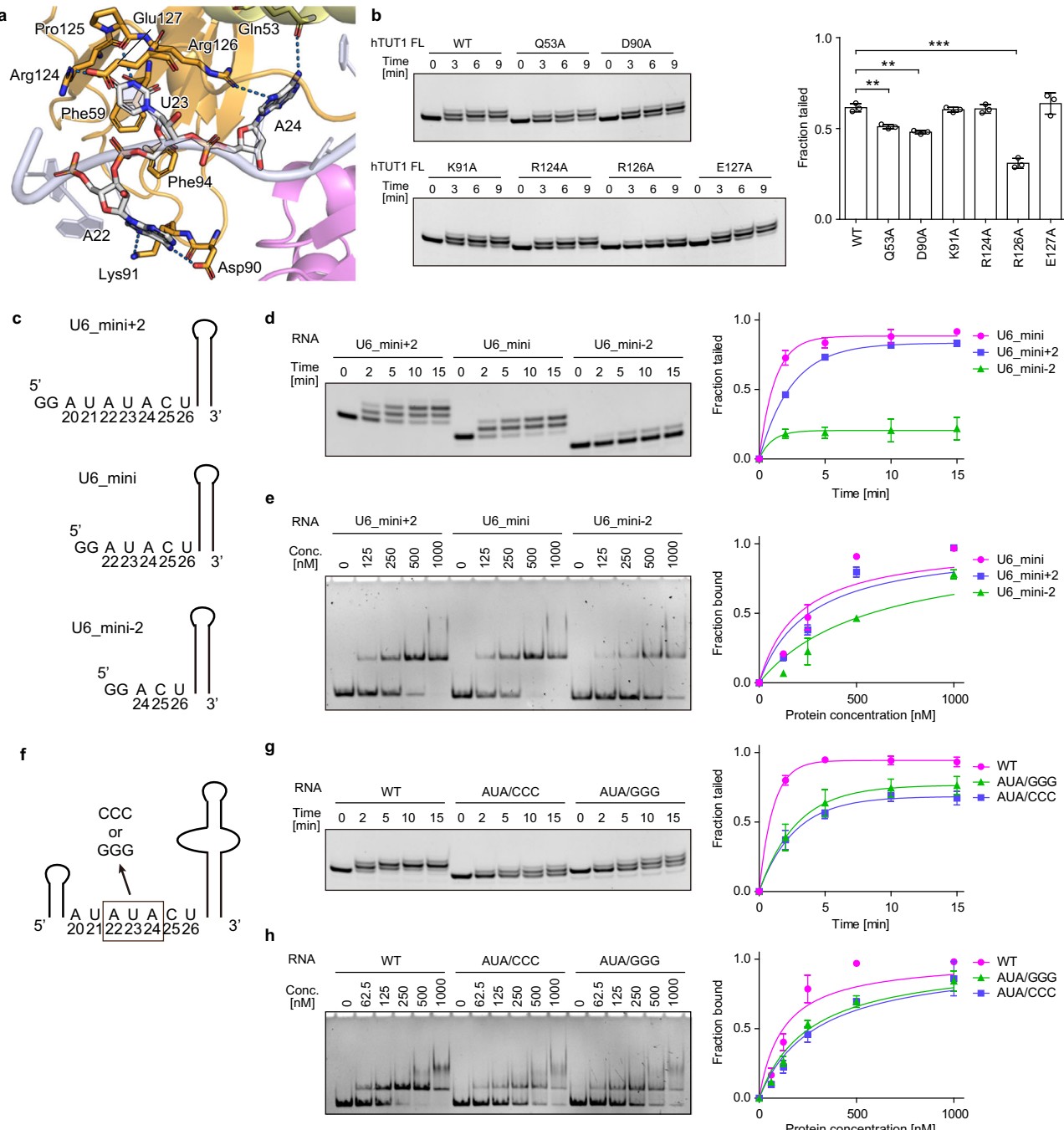

**Fig. 3 | Recognition of the AUA motif in U6 snRNA by the ZF-RRM of hTUT1.**
**a** Recognition of the AUA-motif (A22U23A24) by the N-terminal ZF and RRM of hTUT1. **b** Oligouridylation of U6 snRNA with 4Us by hTUT1_FL and its variants. The U6 snRNA (200 nM) was incubated with 10 nM hTUT1_FL and its variants in the presence of 1 mM UTP. The tailed fractions at 3 min reaction time points in the gels (left) were quantified and presented as mean values ± SD (right). The error bars in the graph represent the SDs of three independent experiments (*n* = 3), and asterisks (*) indicate *p*-values (two-tailed Welch's *t*-test, ***p* < 0.01, ****p* < 0.001). The exact *p*-values are as follows; *p* = 0.0056 (Q53A), *p* = 0.0024 (D90A) and *p* = 0.00011 (R126A). Source data are provided as a Source Data file. **c** Schematic presentation of U6_mini with four 3'-Us and its variants (U6_mini+2 and U6_mini−2) used for oligouridylation by hTUT1_FL. **d** Uridylation of U6_mini and its variants in (**c**) by hTUT1_FL. RNA (200 nM of U6_mini or its variants) was incubated with 20 nM

hTUT1_FL in the presence of 1 mM UTP. The tailed fractions in the gels (left) were quantified (right). **e** Gel shifts of U6_mini and its variants in (**c**) by hTUT1_ZF-RRM (residues 1–140). RNAs (100 nM) were incubated with various amounts of hTUT1_ZF-RRM (0–1000 nM). The fractions of the shifted RNA in the gels (left) were quantified (right). **f** Schematic presentation of U6 with four 3'-Us and its variants used for oligouridylation by hTUT1_FL. (**g**) Uridylation of U6 snRNA and its variants in (**f**) by hTUT1_FL. RNA (200 nM of U6 snRNA or its variants) was incubated with 20 nM hTUT1_FL in the presence of 1 mM UTP. The tailed fractions in the gels (left) were quantified (right). **h** Gel shifts of U6 snRNA and its variants in (**f**) by hTUT1_ZF-RRM, as in (**e**). The fractions of shifted RNA in the gels (left) were quantified (right). The experiments in (**d**), (**e**), (**g**), and (**h**) were performed three times (*n* = 3), and the data are presented as mean values ± SD. Source data are provided as a Source Data file.

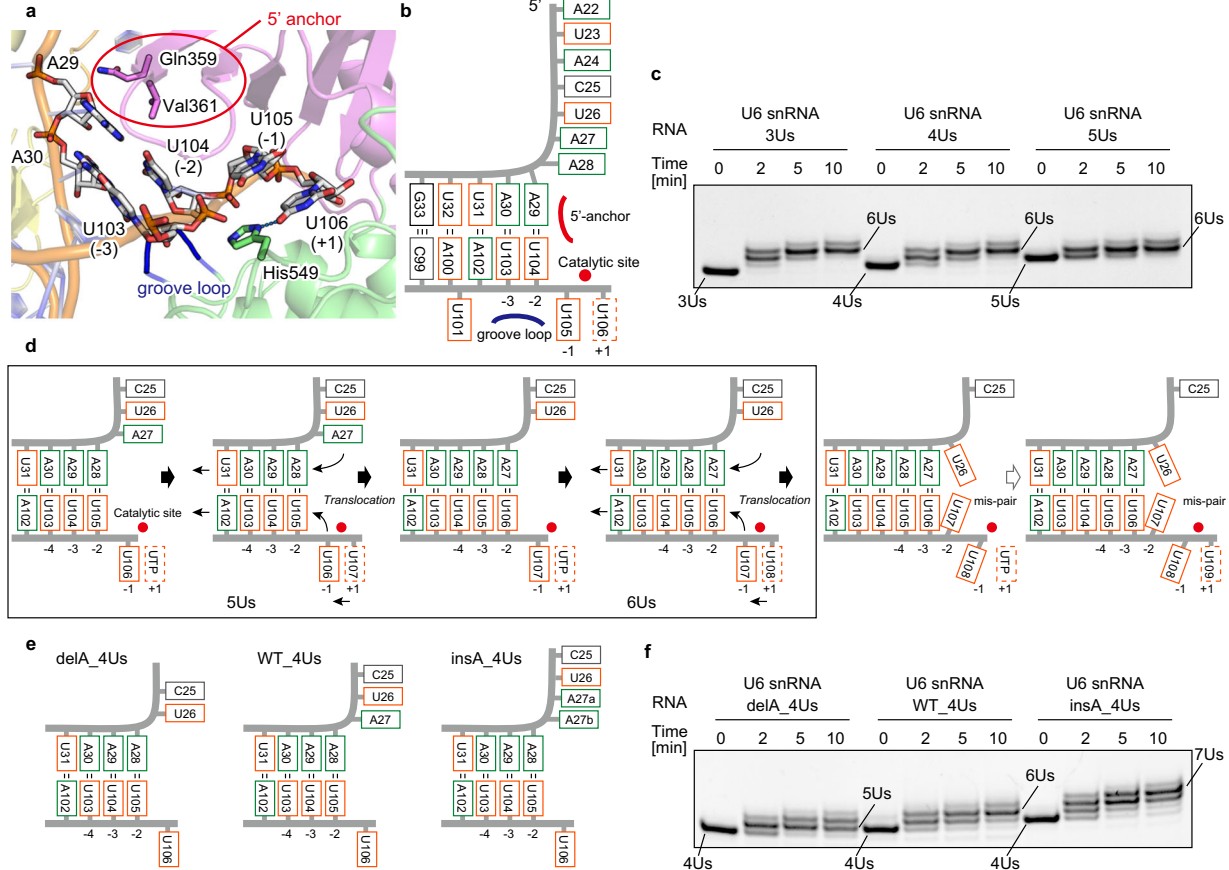

**Fig. 4 | Regulation of U6 snRNA oligouridylation by hTUT1. a** Catalytic core structure of the hTUT1_ΔC-U6_mini complex. **b** Schematic diagram of the secondary structure of U6_mini in the structure of the hTUT1_ΔC−U6_mini complex, representing the post-4U addition state. U106 and U105 are located at the +1 and −1 positions in the catalytic pocket of hTUT1, respectively. **c** Oligouridylation of U6 snRNAs with three 3'-Us (3 Us: U103-U105), four Us (4 Us: U103-U106), and five Us (5 Us: U103-U107) by hTUT1_FL. hTUT1_FL efficiently synthesizes U6 snRNA with up to six 3'-Us (6 Us: U103-U108). U6 snRNAs (200 nM) with three to five 3'-Us were incubated with 20 nM hTUT1_FL in the presence of 1 mM UTP. **d** Possible mechanism of oligouridylation by hTUT1.

Oligouridylation proceeds efficiently until the adenine tract (four adenines: A27- A30) in the telestem base pairs with the 3'-oligouridylated tracts (U103−U106). **e** Schematic diagrams of U6 snRNA variants, delA_4Us and insA_4Us, as well as WT, used in the oligouridylation by hTUT1_FL. One adenine is deleted from A27-A30 (delA_4 Us) and one adenine is inserted in A27-A30 (insA_4 Us). **f** Oligouridylation of U6 snRNA variants in (**e**) (delA_4Us and insA_4Us) by hTUT1_FL. U6 snRNAs (200 nM) with four 3'-Us and its variants were incubated with 20 nM hTUT1_FL in the presence of 1 mM UTP. The experiments in (**c**) and (**f**) were performed three times (n = 3) and the representative gel images are shown. Source data are provided as a Source Data file.

inefficient uridylation after U108 addition (Fig. 4c). The U6 snRNA variants with nucleotide substitutions in the telestem also support the proposal that the oligouridylation of U6 snRNA depends on the base-pairings between nucleotides at positions 27−30 and those at positions 103−106 in the telestem of U6 snRNA (Supplementary Fig. 10). Altogether, the oligouridylation by hTUT1 is controlled and ensured by the intrinsic sequence within the U6 snRNA.

## Discussion

In this study, we determined the crystal structure of a short human TUT1 lacking the C-terminal KA-1, complexed with a short U6 snRNA consisting of a 5'-single strand AUA motif and a 3'-telestem (Fig. 1). The N-terminal ZF-RRM and the palm clamp the single-stranded region containing the AUA motif of U6 snRNA between the 5'-short stem and 3'-telestem, and the telestem is held by the ZF and the fingers (Fig. 1d, e, Supplementary Figs. 4, 5). The single-stranded AUA motif is specifically recognized by the ZF-RRM (Fig. 3a, Supplementary Fig. 5). As a result, the 3'-end of the U6 snRNA is placed in the catalytic cleft between the palm and fingers for the oligouridylation of U6 snRNA. A recent systematic analysis of RNA-motifs recognized by RNA-binding proteins showed that hTUT1 interacts with the AUAcu motif in RNA[29], and the motif sequence matches that between the 5'-short stem and 3'-

telestem of U6 snRNA (Fig. 1b). These structural features effectively explain the previous data showing that the N-terminal ZF-RRM not only increased the affinity of hTUT1 for U6 snRNA but also assisted in the proper positioning of the 3'-end of U6 snRNA in the catalytic site for catalysis[22]. Thus, the recognition of the specific sequence and structure of U6 snRNA by the multiple domains of hTUT1 facilitates the efficient oligouridylation of U6 snRNA.

There are three human terminal uridylyltransferases (TUTases): TUT1, TUT4 and TUT7. While TUT1 and TUT4/7 belong to the TENT family, their domain compositions and substrate RNAs are different[18,23,30] (Fig. 5a). TUT4 and TUT7 uridylate precursor miRNAs, mature miRNAs and mRNAs with short A-tails[31–35]. The uridylation of precursor let-7 miRNAs regulates the expression of their mature counterparts[36–42]. The mechanism of U6 snRNA oligouridylation by TUT1 (Fig. 5b) is distinct from that of pre-let7 uridylation by TUT4/7[43]. TUT4/7 itself does not have oligouridylation activity with pre-let7 (Fig. 5c). In the absence of the Lin28 protein, TUT4/7 monouridylates group II pre-let7 with one 3'-nucleotide overhang[41]. However, in the presence of the Lin28 protein, TUT4/7 oligouridylates pre-let7. Lin28 specifically recognizes the conserved GGAG motif in the terminal loop of pre-let7 miRNAs[44,45] and recruits the N-terminal Lin28-interacting module (LIM) of TUT4/7 to pre-let7[23,40,44–46]. The stable

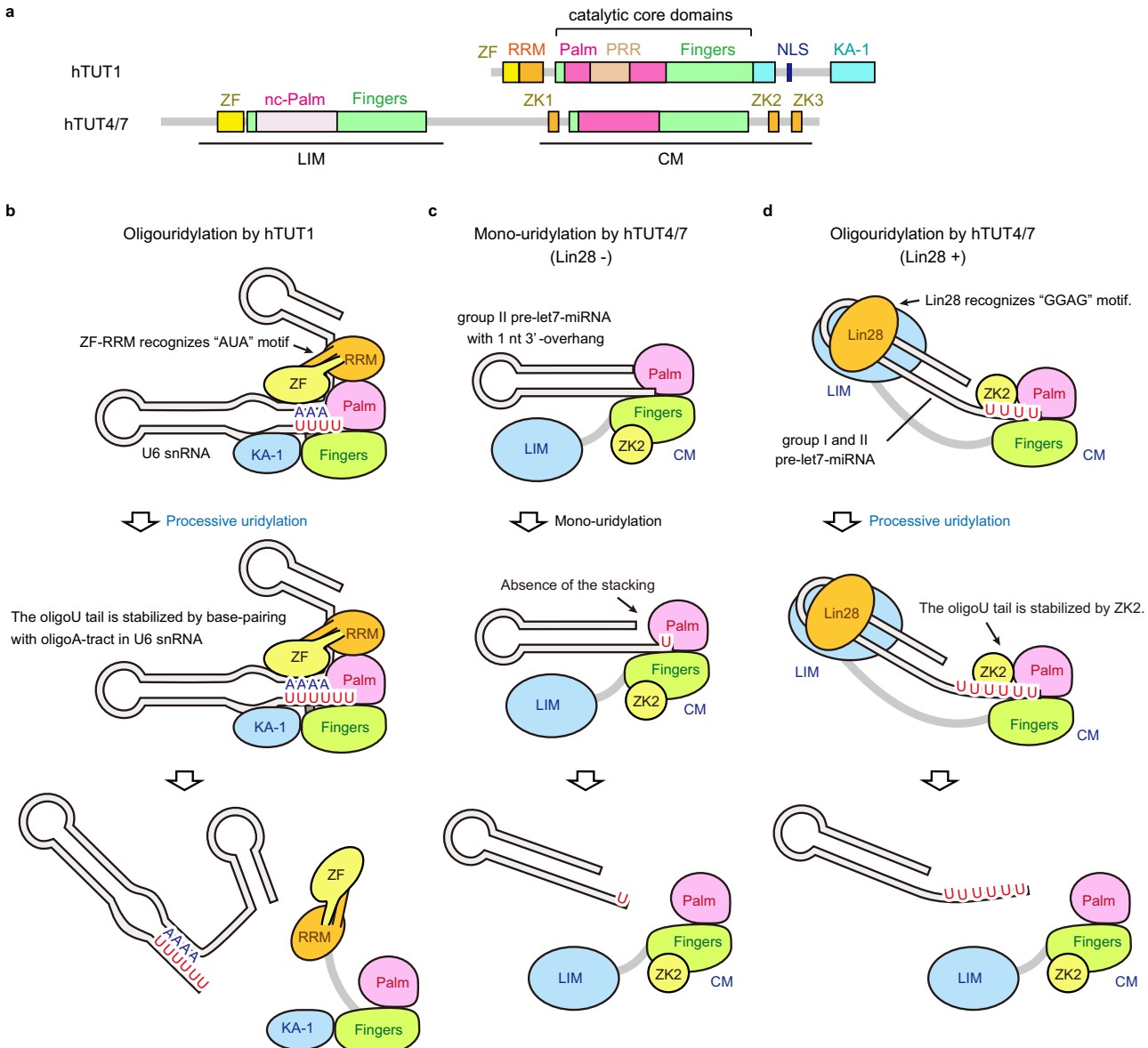

**Fig. 5 | Comparison of the uridylation mechanisms between hTUT1 and hTUT4/7. a** Schematic diagram of the domain compositions of hTUT1 (above) and hTUT4/7 (below). LIM (Lin28 interacting module) and CM (catalytic module) of TUT4/7 are underlined below the diagram. ZKs (zinc knuckles: ZK1-3) are colored orange. **b** Mechanism of U6 snRNA oligouridylation by hTUT1. **c** Mechanism of the group II pre-let7 monouridylation with one overhanging 3'-nucleotide by TUT4/7 in the absence of Lin28 protein. **d** Mechanism of pre-let7 oligouridylation in the presence of the Lin28 protein.

ternary complex facilitates the processive oligouridylation of pre-let7 by the C-terminal catalytic module (CM) of TUT4/7 (Fig. 5d). The oligouridylation proceeds in the CM with the assistance of zinc knuckle 2 (ZK2) in the CM of TUT4/7, which interacts with the uridine at the −2 position of the oligouridylated tail, thus stabilizing it[43] (Fig. 5d). In contrast, the oligouridylation of U6 snRNA proceeds by hTUT1 alone and does not require additional RNA-binding proteins, and there is no domain equivalent to ZK2 in TUT4/7 (Fig. 5a). hTUT1 clamps the single-stranded region containing the AUA motif between the 5'-short stem and 3'-telestem and forms the stable hTUT1-U6 snRNA complex[10]. The oligouridylation by hTUT1 is regulated by the adenine tract in the telestem (Fig. 4e, f, Supplementary Fig. 10). Oligouridylation proceeds until the uridine at the −2 position (U106) forms a base pair with A27 in the adenine tract in the telestem (Fig. 4d). hTUT1 utilizes the intrinsic specific sequence, the adenine tract, to ensure the uridine incorporations onto the 3'-end of U6 snRNA, instead of using a specific uridine binding domain such as

ZK2 in TUT4/7 (Fig. 5d). Recently, it has been reported that AtURT1, an *Arabidopsis thaliana* TUTase, adds two uridines to the 3'-end of poly(A). Residues L527 and Y592 in URT1 contribute to its preference for purine over pyrimidine at the -2 position (the 3'-priming position corresponds to -1) of single-stranded RNA[47]. This preference enables URT1 to control the optimal number of uridines and URT1 added two uridines to the 3' extremity of polyA. L527 and Y592 of AtURT1 corresponds to V361 and G426 of hTUT1, respectively. V361 is located at the 5'-anchor of hTUT1 and stacks with the bases at the top of the stem (Fig. 4a). G426 is in groove loop which interacts with the minor groove of the stem (Fig. 4a, Supplementary Fig. 5g). Thus, the mechanisms of controlling the number of uridines added to the 3'-end of RNAs are different between AtURT1 and hTUT1.

In sum, the intrinsic sequence and structure of U6 snRNA and the recognition of the U6 snRNA structure by the multi-domain hTUT1 protein together regulate and ensure the oligouridylation of U6 snRNA, which is essential for the splicing of pre-mRNAs.

# Methods

### Plasmids
The synthetic DNA encoding human TUT1 (hTUT1) and its variants were purchased from Eurofins Genomics (Japan). The nucleotide sequences of the synthetic hTUT1 gene and its variants are shown in Supplementary Table 2. The DNA fragments encoding the full-length hTUT1 and its variants were cloned between the NdeI and XhoI sites of the pET22b vector (Merck Millipore, Japan; 69744-3CN). The mutations were introduced by the inverse or overlap PCR method. The nucleotide sequences of the RNAs used in this study are listed in Supplementary Table 3. oligonucleotide sequences of the primers used are listed in Supplementary Table 4.

### Expression and purification of recombinant proteins
*E. coli* BL21(DE3) (Novagen, Japan; 69450-3CN) was transformed by each plasmid, and grown at 37 °C until the $A_{600}$ reached 1.0. The expression of the hTUT1 protein or its variants was induced by adding 0.1 mM isopropyl-β-D-thiogalactopyranoside (IPTG) and continuing the culture at 18 °C for 16 h. The cells were harvested and lysed in buffer, containing 20 mM Tris-HCl, pH 7.0, 500 mM NaCl, 10 mM β-mercaptoethanol, 20 mM imidazole, 0.1 mM phenylmethylsulfonyl fluoride (PMSF) and 5% (v/v) glycerol. The proteins were first purified on a Ni-NTA agarose column (QIAGEN, Japan; 30210), followed by a HiTrap Heparin column (GE Healthcare, Japan; 17-0407-01). Finally, the proteins were purified on a HiLoad 16/60 Superdex 200 column (GE Healthcare, Japan; 17-1069-01), in buffer containing 20 mM Tris-HCl, pH 7.0, 300 mM NaCl, and 10 mM β-mercaptoethanol. The purified proteins were concentrated and stored at −80 °C.

### RNA preparations
Synthetic human U6 snRNA and its variants were synthesized by T7 RNA polymerase using the corresponding DNA fragments as templates. To prepare U6 snRNA (or its variants) with the homogeneous 3′-end, the DNA fragment encoding the U6 snRNA gene (or its variants) carrying the T7 promoter and the HDV ribozyme sequence upstream and downstream of the U6 snRNA gene (or its variants) sequence, respectively[48], was used as the template. After in vitro transcription, the RNA was phenol-chloroform-extracted and dissolved in buffer containing 10 mM Tris-HCl, pH 7.0, and 20 mM MgCl$_2$. The RNA solution was subjected to 15 cycles of incubations (at 60 °C for 3 min and 25 °C for 3 min) to self-cleave the RNA by the HDV ribozyme. The 3′-cyclic phosphate of RNA after the HDV self-cleavage was removed by T4 kinase (Takara, Japan; 2021 A), in buffer containing 50 mM MES, pH 5.8, 20 mM MgCl$_2$, and 10 mM DTT. Finally, the RNA was purified by 10% (w/v) polyacrylamide gel electrophoresis under denaturing conditions. The synthetic oligonucleotide sequences of the U6 snRNA gene and its variants used for in vitro transcription are listed in Supplementary Table 2.

### Crystallization and structural determination
The crystal used for the structural determination was generated by the sitting drop vapor diffusion method at 4 °C. Before crystallization, 25 μM protein and 30 μM RNA were mixed in a solution containing 10 mM Tris-HCl, pH 7.0, 150 mM NaCl, 5 mM β-mercaptoethanol, 10 mM DTT, and 50 mM zinc acetate. A 200 nL portion of the protein-RNA solution was mixed with 100 nL of the reservoir solution, containing 18–20% (w/v) PEG3350, 100 mM Bis-Tris, pH 6.4–6.5, 2% (v/v) Tacsimate, pH 6.0 (Hampton; HR2-827), and 8% (v/v) acetonitrile. The data set was collected at beamline 17 A at the Photon Factory at KEK, Japan. The crystals were flash-cooled with the reservoir solution supplemented with 30% (v/v) PEG400. The dataset was indexed, integrated, and scaled with XDS[49]. The initial phase was determined by molecular replacement with Phaser[50], using the catalytic core structures of hTUT1 (PDB IDs: 5WU1 and 5WU6)[22]. The structure was refined with phenix.refine[51] and manually modified with Coot[52]. All figures were prepared with PyMOL (http://www.pymol.org).

### Uridylation assays
A reaction mixture (30 or 40 μL), containing 50 mM Tris-HCl, pH 8.5, 100 mM NaCl, 10 mM MgCl$_2$, 10 mM β-mercaptoethanol, 1 mM UTP, and the indicated concentrations of RNA transcript and TUT1 (or its variants), was incubated at 37 °C. At the indicated time points, a 5 μL portion of the reaction mixture was withdrawn, and the reactions were stopped. The RNAs were separated by 10% (w/v) polyacrylamide gel electrophoresis under denaturing conditions and stained with ethidium bromide. The band intensities were quantified with Gel Doc EZ, using the Image Lab software (Bio-Rad, Japan).

### Gel-shift assay
RNAs (100 nM U6_mini and its variants or 50 nM U6 snRNA and its variants) were mixed with various concentrations (0–1000 nM) of hTUT1_ZF-RRM (residues 1-140) or its variants in 10 μL of buffer, containing 50 mM Tris-Cl, pH 8.5, 100 mM NaCl, 10 mM MgCl$_2$, 10 mM β-mercaptoethanol, and 10% (v/v) glycerol. The mixture was incubated at room temperature for 15 min. The RNAs were separated by 6% (w/v) or 8% (w/v) polyacrylamide gel electrophoresis under native conditions at room temperature, and the gels were stained with ethidium bromide. The shifted band intensities were quantified with Gel Doc EZ, using the Image Lab software (Bio-Rad, Japan).

### Reporting summary
Further information on research design is available in the Nature Portfolio Reporting Summary linked to this article.

## Data availability
The data supporting the findings of this study are available from the corresponding authors upon reasonable request. The coordinates and structure factors of hTUT1_ΔC-U6_mini have been deposited in the Protein Data Bank, under the accession code 8IDF. Source data for the figures and supplementary figures are provided as a Source Data file. Source data are provided with this paper.

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

## Acknowledgements

We thank the beamline staff of BL-17A (KEK, Tsukuba) for technical assistance during data collection. This work was supported by grants from the Funding Program for Next Generation World-Leading Researchers of JSPS [grant number LS135 to K.T.], Grants-in-Aid for Scientific Research (A) [grant numbers 23H00368, 18H03980 and 26251009 to K.T.], a Grant-in-Aid for Early-Career Scientists [grant number 19K16053 to S.Y.] from JSPS, and a Grant-in-Aid for Scientific Research on Innovative Areas from the Ministry of Education, Culture, Sports, Science, and Technology of Japan [grant number 26113002 to K.T.]. K.T. was also supported by grants from the Uehara Memorial

Foundation, Japan, the Terumo Foundation for Life Science and Art, and the Princess Takamatsu Cancer Research Fund.

## Author contributions

K.T. planned and designed the research. S.Y. performed structural and biochemical analyses. S.Y. and K.T. analyzed the data and wrote the paper.

## Competing interests

The authors declare no competing interests.
