## [Peer Review File · Nature Communications]

REVIEWER COMMENTS

Reviewer #1 (Remarks to the Author):

The current paper reports on the crystal structure of a truncated (lacking C-terminal Ka-1 domain) terminal uridylyltransferase TUT1 bound to a truncated U6 snRNA (lacking 5'-short stem and 3'-ILS segments). The structure is quite striking in identifying key protein-RNA contacts important for the addition of uridines to the 3'-end of U6 RNA. The paper expands on earlier research by the same group on the structure of Tut1 (lacking N-terminal ZF and RRM domains) bound to UTP and ATP reported in 2017. Key insights in the current structure are the specific clamping of an AUA motif by the ZF-RRM and palm domains, the engagement of the ZF and finger domains in holding the telestem segment of the RNA, thereby positioning the 3'-end of the U6 snRNA in the catalytic pocket of the palm domain for uridylation at its 3'-end. The authors perform biochemical experiments to establish that addition of uridines to the 3'-end of U6 snRNA is dependent on pairing with an A tract positioned at the 5'-end of the telestem. The results represent an importance advance in the field.

I recommend publication once the authors have addressed an issue of concern related to the crystallographic analysis of the complex.

Concern: The Rsym value is too high while the redundancy value is also outside the normal range in the data statistics listed in Table S1. This may require further refinement to correct for these discrepancies. The Supp Table should also include information on a Ramachandran plot, rotamer outliers and clash score. In addition, an RNA omit map should be included as a Supplementary Figure. There also was no Validation Report provided for inspection.

The in vitro uridylation assay reported in this paper suggests that TUT1 has a preference for di-uridylation activity towards U6 snRNA. A recent paper (Hu et al. *Nucleic Acids Res.* 50, 10614-10625, 2022) has explained the mechanism of di-uridylation activity of URT1, an Arabidopsis Tutase, using L527 and Y592 to discriminate between purine and pyrimidine. The authors should consider whether the equivalent amino acids in TUT1 function in the same way, and recommend a discussion of this issue in a revised version.

Reviewer #2 (Remarks to the Author):

In the manuscript “Mechanism of U6 snRNA oligouridylation by human TUT1” by Yamashita et al, the authors present a crystal structure of U6 in complex with TUT1 to investigate the mechanism of oligouridylation.

While the complex structure is useful, the mechanistic insight that it shows is somewhat limited. The structure data is at a modest resolution, and the contacts between TUT1 and U6 ends seem to be particularly less resolved, making the interesting region for uridylation even more difficult to interpret. The authors raise an interesting comparison with other TUTases, but the focus on the kinetic regulation is also not thoroughly addressed with experiments. Here are some suggestions to help improve the manuscript:

1. There is a lot of discussion of enzyme kinetics, and “regulation of the kinetics”. When the authors say “faster”, could they clarify which parameter they are referring to? Although time courses are shown, they are in single concentrations, and from the binding assays does not seem like where the turnover is limited.
2. For the protein mutants (Fig 3b), can the authors show that the affinity for the AUA motif RNA is altered? And does the affinity change get affected by changes to the AUA motif?
3. “Thus, we designed a shorter hTUT1 and a shorter U6 snRNA for the crystallization of the hTUT1-U6 snRNA complex” : Please specify the constructs.
4. How do the authors know that the several mutations they introduced during crystallization do not affect the enzyme activity?
5. Most of the structures of individual components were previously determined already at higher resolution. Taking advantage of these available structures, can the authors improve the complex structure model quality? RFree is on the higher side than average, and RSRZ could improve even at this resolution.

Reviewer #1

The current paper reports on the crystal structure of a truncated (lacking C-terminal Ka-1 domain) terminal uridylyltransferase TUT1 bound to a truncated U6 snRNA (lacking 5'-short stem and 3'-ILS segments). The structure is quite striking in identifying key protein-RNA contacts important for the addition of uridines to the 3'-end of U6 RNA. The paper expands on earlier research by the same group on the structure of Tut1 (lacking N-terminal ZF and RRM domains) bound to UTP and ATP reported in 2017. Key insights in the current structure are the specific clamping of an AUA motif by the ZF-RRM and palm domains, the engagement of the ZF and finger domains in holding the telestem segment of the RNA, thereby positioning the 3'-end of the U6 snRNA in the catalytic pocket of the palm domain for uridylation at its 3'-end. The authors perform biochemical experiments to establish that addition of uridines to the 3'-end of U6 snRNA is dependent on pairing with an A tract positioned at the 5'-end of the telestem. The results represent an importance advance in the field.

I recommend publication once the authors have addressed an issue of concern related to the crystallographic analysis of the complex.

Our response.

Thank you for your favorable comments on our manuscript. According to the comments and suggestions, we revised the manuscript.

1) Concern: The Rsym value is too high while the redundancy value is also outside the normal range in the data statistics listed in Table S1. This may require further refinement to correct for these discrepancies. The Supp Table should also include information on a Ramachandran plot, rotomer outliers and clash score. In addition, an RNA omit map should be included as a Supplementary Figure. There also was no Validation Report provided for inspection.

Our response.

Thank you for your comments.

i) Higher Rsym value.

We agree that Rsym value is relatively high. We made every effort to improve the quality of the statistics of the refinements. Although we collected several data sets, all the data showed the relatively higher Rsym and Rmeas values. We integrated and scaled the data sets considering that the crystal might belong to other space group, such as P6, P312, P321, P222. However, the Rsym and Rmeas values were still high. We assume that it might arise from some heterogeneity of crystal packing, because the crystals were dehydrated during cryoprotection by PEG400, and the unit cell of the crystals become smaller.

ii) Redundancy.

The crystal belongs to the space group P6522 (12 times symmetry). We collected the diffraction data for 720 degree (Passing the Ewald sphere 8 times at maximum). Thus, the obtained high redundancy (72.2) is reasonable.

iii) According to the suggestion, we included the information on a Ramachandran plot, rotomer outliers and clash score in the revised manuscript (Supplementary Table 1).

- iv) According to the suggestions we provided RNA omit maps in the revised manuscript (Supplementary Fig. 3). These electron density maps support the presented structure.
- v) According to the suggestions we provided the Validation Reports.

2) The in vitro uridylation assay reported in this paper suggests that TUT1 has a preference for di-uridylation activity towards U6 snRNA. A recent paper (Hu et al. Nucleic Acids Res. 50, 10614-10625, 2022) has explained the mechanism of di-uridylation activity of URT1, an Arabidopsis Tutase, using L527 and Y592 to discriminate between purine and pyrimidine. The authors should consider whether the equivalent amino acids in TUT1 function in the same way, and recommend a discussion of this issue in a revised version.

Our response.

According to the suggestion, we described the comparison between *Arabidopsis TUTase* URT1 and TUT1 as follows in the discussion section.

“Recently, it has been reported that AtURT1, an *Arabidopsis thaliana* TUTase, adds two uridines to the 3'-end of poly(A). Additionally, residues L527 and Y592 in URT1 contribute to its preference for purine over pyrimidine at the -2 position (the 3'-priming position corresponds to -1) of single-stranded RNA. This preference enables URT1 to control the optimal number of uridines and URT1 added two uridines to the 3' extremity of polyA. L527 and Y592 of AtURT1 corresponds to V361 and G426 of hTUT1, respectively. V361 is located at the 5'-anchor of hTUT1 and stacks with the bases at the top of the stem (**Fig. 4a**). G426 is in groove loop which interacts with the minor groove of the stem (**Fig. 4a, Supplementary Fig. 5g**). Thus, the mechanisms of controlling the number of uridines added to the 3'-end of RNAs are different between AtURT1 and hTUT1. ”

Reviewer #2

In the manuscript “Mechanism of U6 snRNA oligouridylation by human TUT1” by Yamashita et al, the authors present a crystal structure of U6 in complex with TUT1 to investigate the mechanism of oligouridylation.

While the complex structure is useful, the mechanistic insight that it shows is somewhat limited. The structure data is at a modest resolution, and the contacts between TUT1 and U6 ends seem to be particularly less resolved, making the interesting region for uridylation even more difficult to interpret. The authors raise an interesting comparison with other TUTases, but the focus on the kinetic regulation is also not thoroughly addressed with experiments. Here are some suggestions to help improve the manuscript:

Our response.

Thank you for your valuable and constructive comments and suggestions for improving our manuscript. According to the comments and suggestions, we provided additional experimental data and supplementary figures, and revised the manuscript.

1. There is a lot of discussion of enzyme kinetics, and “regulation of the kinetics”. When the authors

say “faster”, could they clarify which parameter they are referring to? Although time courses are shown, they are in single concentrations, and from the binding assays does not seem like where the turnover is limited.

Our response.

Thank you for your comments.

We agree that the wording of kinetics is very confusing and ambiguous in the text.

Thus, we avoid using the word, kinetics in the revised manuscript and carefully revised the text.

2. For the protein mutants (Fig 3b), can the authors show that the affinity for the AUA motif RNA is altered? And does the affinity change get affected by changes to the AUA motif?

Our response.

Thank you for your comments.

According to the comments, we tested the RNA binding of the TUT1 mutants in the revised manuscript (Supplementary Figure 8 in the revised manuscript).

The results showed that Q53A/R126A and F59A/F94A mutations in ZF-RMM decrease the affinity for U6 snRNA with A22U23A24 motif.

3. “Thus, we designed a shorter hTUT1 and a shorter U6 snRNA for the crystallization of the hTUT1-U6 snRNA complex” : Please specify the constructs.

Our response.

We clearly described the constructs in text as follows.

“Therefore, for a short hTUT1, we designed the hTUT1 lacking the PRR and the KA-1, hereafter termed hTUT1_ΔC (**Fig. 1a**). For a shorter U6 snRNA, we designed U6 snRNA lacking the 5'-short stem and the ISL and possessing four 3'-Us (U103-U106), hereafter termed U6_mini (**Fig. 1b**)”

4. How do the authors know that the several mutations they introduced during crystallization do not affect the enzyme activity?

Our response.

According to the suggestions, we tested the uridylation of U6 snRNA by TUT1 protein with several mutations introduced for crystallization, in the revised manuscript. These mutations do not affect the uridylation of U6 snRNA. (Supplementary Figure 2 in the revised manuscript).

5. Most of the structures of individual components were previously determined already at higher resolution. Taking advantage of these available structures, can the authors improve the complex structure model quality? RFree is on the higher side than average, and RSRZ could improve even at this resolution.

Our response.

i) We used the available structures for the initial models and refined the complex structure.

ii) We agree that the R_{free} and RSRZ values are on the higher side. As responded to the comments from Reviewer #1, we made every effort to improve the quality of the structure. We tested the various refinement conditions and presented the best one. The slightly worse statistics might arise from the heterogeneity in the crystal packing.

REVIEWERS' COMMENTS

Reviewer #1 (Remarks to the Author):

The authors have addressed my concerns and the paper is ready for publication.

Reviewer #2 (Remarks to the Author):

If this is the best model the authors can build, the manuscript is ready for publication despite the subpar statistics. There is a typo in the crystallography table: "Outliners" should be "Outliers".

Reviewer #1

The authors have addressed my concerns and the paper is ready for publication.

Our response

Thank you for your supportive comments and constructive comments on our manuscript.

Reviewer #2

If this is the best model the authors can build, the manuscript is ready for publication despite the subpar statistics. There is a typo in the crystallography table: "Outliners" should be "Outliers".

Our response

Thank you for your supportive comments and constructive comments on our manuscript.

We corrected the typo in the table.